# Optical Flow Sensor with Fluorescent-Conjugated Hyperelastic Pillar: A Biomimetic Approach

**DOI:** 10.3390/biomimetics9120721

**Published:** 2024-11-22

**Authors:** Dongmin Seo, Seungmin Yoon, Jaemin Park, Sangwon Lee, Seungoh Han, Sung-Hoon Byun, Sangwoo Oh

**Affiliations:** 1Department of Electrical and Electronic Engineering, Semyung University, Jecheon 27136, Republic of Korea; dseo@semyung.ac.kr; 2Department of Electrical Engineering, Semyung University, Jecheon 27136, Republic of Korea; ysmin5792@gmail.com (S.Y.); jm77788828@gmail.com (J.P.); sw2796@naver.com (S.L.); 3Department of Robotics, Hoseo University, Asan 31499, Republic of Korea; sohan@hoseo.edu; 4Ocean and Maritime Digital Technology Research Division, Korea Research Institute of Ships and Ocean Engineering, Daejeon 34103, Republic of Korea; byunsh@kriso.re.kr

**Keywords:** bio-inspired sensor, optical flow sensing, underwater velocity measurement, fluorescent-conjugated hyperelastic pillar, neuromast mimicking

## Abstract

Although the Doppler velocity log is widely applied to measure underwater fluid flow, it requires high power and is inappropriate for measuring low flow velocity. This study proposes a fluid flow sensor that utilizes optical flow sensing. The proposed sensor mimics the neuromast of a fish by attaching a phosphor to two pillar structures (A and B) produced using ethylene propylene diene monomer rubber. The optical signal emitted by the phosphor is measured using a camera. An experiment was conducted to apply an external force to the reactive part using a push–pull force gauge sensor to confirm the performance of the proposed sensor. The optical signal emitted by the phosphor was obtained using an image sensor, and a quantitative value was calculated using image analysis. A simulation environment was constructed to analyze the flow field and derive the relationship between the flow rate and velocity. The physical properties of the pillar were derived from hysteresis measurement results, and the error was minimized when pillar types A and B were utilized within the ranges of 0–0.1 N and 0–2 N, respectively. A difference in the elastic recovery characteristics was observed; this difference was linear based on the shape of the pillar, and improvement rates of 99.585% and 99.825% were achieved for types A and B, respectively. The proposed sensor can help obtain important information, such as precise flow velocity measurements in the near field, to precisely navigate underwater unmanned undersea vehicles and precisely control underwater robots after applying the technology to the surface of various underwater systems.

## 1. Introduction

Understanding the marine environment is crucial in terms of climate change, ecosystem protection, and marine resource management [1,2]. Monitoring the marine environment is essential for accurately understanding it, and various sensors need to be actively developed [3,4]. Among them, the accurate measurement of underwater fluid flow provides important data for assessing the stability of marine structures, managing marine ecosystems and aquaculture farms, navigating unmanned underwater vehicles, and designing tidal power plants [5,6,7,8]. Therefore, the development of reliable flow velocity measurement technology in the marine environment is crucial for both academia and industry.

The hot wire anemometer and Doppler velocity log are the most frequently used equipment for measuring flow velocity. The former calculates flow velocity by passing a current through a thin resistance wire and measuring the change in the calorific or resistance value in line with flow velocity [9]. However, this can be affected by the thermal conductivity of water, which can lead to inaccuracies. Therefore, this sensor is recommended for use in indoor environments, where parameters such as wind speed, air volume, and air temperature can be easily measured. The Doppler velocity log utilizes the Doppler effect to measure the flow of underwater fluids [10,11]. The Doppler effect causes the light irradiated on a moving object to scatter, thereby causing a frequency shift proportional to the velocity of the object. The Doppler velocity log measures flow velocity by detecting the frequency shift of the light scattered by the particles in the fluid. However, the Doppler velocity log is inappropriate for measuring flow velocity in inaccessible locations because of its lower sensitivity under slow flow, dependence on the particle concentration in the fluid, and high power consumption. Recent advancements in optical flow sensing have introduced novel approaches to water flow monitoring. For example, a recent study demonstrated a low-cost optical sensor that captures changes in flow velocities by analyzing changes in optical properties within a confined tube, offering an alternative to traditional high-power methods [12]. This approach, while effective in controlled environments, requires a transparent tube for flow visualization. By contrast, our biomimetic sensor is designed for deployment in open water, enabling in situ marine flow sensing without the constraints of confined structures.

To address these problems, researchers have been exploring biomimetic principles for underwater flow velocity measurements. Recent research has explored the development of hair cell-inspired biomimetic mechanoreceptors for underwater flow and acoustic sensing, with potential applications in artificial lateral line (ALL) systems for autonomous underwater vehicles and biosensing [13]. Additionally, ALL systems inspired by the fish lateral line have focused on replicating neuromast structures and enhancing flow perception for underwater applications through biomimetic mechanisms [14]. These advanced ALL sensors have shown the potential for improving underwater robot navigation by enhancing flow sensing, environmental interaction, and control systems, thereby contributing to autonomy in aquatic environments [15]. Furthermore, investigations into the “morphological intelligence” of biological flow sensors, such as fish lateral lines and mammalian whiskers, have revealed key design principles, including flow stimulus enhancement, noise reduction, and nonlinear sensitivity, which can inform the development of advanced biomimetic flow sensors [16]. Furthermore, beyond aquatic organisms, the biomechanics of sensory organs in both animals and plants have inspired the design of soft, flexible mechanical sensors for biomimetic and robotic applications [17].

Neuromasts in the lateral line of fish are a prime example of a biological mechanism for sensing the flow of underwater fluids. The neuromasts can be divided into superficial neuromasts protruding outside and internal canal neuromasts. Neuromasts are sensitive to fluid flow and enable fish to detect prey, avoid predators, and maintain their position within a school. The fluid flow displaces the cupula of the neuromast, which moves the kinocilia of the hair cells. When the kinocilia move, the sensory cells of the fish generate electrical signals to recognize changes in the environment. Sensors that mimic this biological mechanism replicate the kinocilia of the hair cells by molding cantilever- or pillar-shaped protrusions, and the difference in the displacement of the replicated kinocilia is analyzed using various sensing methods.

Representative measurement methods for biomimetic flow velocity sensors include piezoresistive, piezoelectric, and capacitive methods. Piezoresistive sensors are based on the piezoresistive effect, where the electrical resistance of a material changes in line with mechanical deformation. A piezoresistive sensor is responsible for one axis of a Wheatstone bridge circuit. The imbalance in the bridge circuit attributed to the resistance that changes with the bending of the kinocilium mimic causes a change in the voltage, which is calculated as the flow velocity. Several studies have replicated a kinocilium in the shape of a cantilever using a piezoresistive sensing method [18,19,20], and other studies have replicated a kinocilium in the shape of a pillar [21,22,23]. Piezoelectric sensors are based on the piezoelectric effect, which generates an electrical charge in response to mechanical stress. The piezoelectric effect occurs in materials such as quartz, ceramics, and some polymers, and the charge generated by the bending of a kinocilium mimic is measured and converted into a flow velocity. Some studies have replicated kinocilia in the shape of pillars using the piezoelectric sensing method [24,25], and other studies have focused on replicating the cupula shape of aggregated kinocilia [26]. Capacitive sensors detect flow velocity by measuring the change in capacitance caused by the distance between conductive plates or the changes in dielectric materials. Studies on single-pillar-shaped geometries using capacitive sensing methods [27] and on arranging multiple kinocilia to utilize them as sensors have been reported [28]. However, these sensors are sensitive to changes in ambient temperature and pressure, which limits their application to variable underwater environments.

To address these problems, some researchers developed flow velocity sensors based on optical measurements that are less sensitive to temperature and pressure changes. Optical measurement uses a pillar structure that mimics the shape of a neuromast in fish and measures the change in the optical signal as the pillar moves. Previous studies utilized a sensing mechanism that arranges an illuminant and detector at the top and bottom of the pillar, respectively, by utilizing a pillar made of transparent material as an optical waveguide [29,30,31]. In addition, some studies engraved a Bragg grating inside the pillar and detected the pillar deformation as a wavelength change caused by fluid flow [32,33,34]. However, the optical waveguide method has various limitations. For instance, it can only measure flow velocity but cannot provide directional information, and the optical fiber grating sensor requires additional equipment to analyze the wavelength of the optical signal, rendering the method inappropriate for application to underwater systems with limited energy supply.

Conventional flow velocity sensors require high power consumption, which limits their use in battery-operated or unmanned systems. Therefore, developing sensors with low power consumption is necessary, and performing measurements using a USB camera can be one alternative [35]. This study proposes a sensor that uses a fluorescent-conjugated hyperelastic pillar to detect mechanical deformations caused by fluid flow. When the pillar is bent by fluid flow, the fluorescent signal emitted from phosphor changes because of the transformation of optical waveguides inside the pillar, and the signal is converted into an image captured by a USB camera to calculate the velocity and direction of the fluid. This approach offers several advantages:(1)The room for external light to function as a noise factor can be minimized by designing the sensor system such that it is located inside the underwater unmanned system (only the pillar structure with the phosphor attached is exposed to the exterior).(2)A wide range of flow velocity can be measured, including very low water flow velocity, by modifying the hardness and design of the pillar.(3)The flexibility of the pillar makes it compatible with diverse materials, enabling the creation of complex sensor arrays that mimic biological structures.(4)The sensor system can be designed with low power and developed as an instrument to measure flow velocity in a location that is difficult to access.

In this study, we devised a method to produce two types of pillars with phosphor attached and compared their elastic properties through experiments. Further, a method was proposed to quantitatively analyze the intensity of the fluorescence signal emitted by phosphor, and the change in the fluorescence signal based on the force applied to the pillar was measured. The elastic properties of the two types of pillars were also analyzed. Further, the properties of the pillar material—ethylene propylene diene monomer (EPDM) rubber—were determined using tensile tests, and the results were confirmed by constructing an environment that can simulate pillar displacements in a flow field. Finally, an equation converting the injected flow rate into the corresponding flow velocity was derived. The remainder of this paper is organized as follows: Section 2 details the materials and methods employed in this study, including the sensor design, pillar fabrication process, experimental setup, and simulation environment. Section 3 presents the results of the experimental and simulation studies, encompassing the sensor performance evaluation, fluorescence signal analysis, and mechanical displacement simulations, demonstrating the sensor’s effectiveness. Finally, Section 4 concludes the study by summarizing the key findings, discussing the limitations, and proposing potential directions for future research.

## 2. Materials and Methods

### 2.1. Sensor Design

The superficial neuromast on the lateral line of fish consists of a series of tiny protrusions that are mechanically deformed as they bend by water flow. The mechanical deformation of the cupula composing this structure affects bundles of hair cells, which are converted into electrical signals that can be perceived by the fish.

The sensor structure proposed in this study was produced as a pillar, which is similar to the shape of the protrusion, to mimic this sensing mechanism. Figure 1 illustrates the shape of the superficial neuromast of the fish and sensor structure developed to mimic it. The pillar was fabricated with EPDM rubber that shows excellent flexibility and durability in an underwater environment. An optical sensing method in which a phosphor is attached to the pillar was employed to measure the pillar movement, and the optical signal emitted from the phosphor was measured using a camera. This optical sensing method was not exposed to the external environment of illuminants, cameras, and other structures, thus minimizing the interference from the external environment. The optical signal measured by the camera was converted into an image, which contained two-dimensional position information and the intensity of the optical signal. The degree and direction of the mechanical deformation of the pillar caused by the water flow were analyzed based on this information. The deformation degree suggested that the intensity of the water flow and direction indicate the direction of the water flow. Overall, the proposed water flow measurement sensor with the optical sensing method can simultaneously measure flow velocity and direction.

### 2.2. Pillar Manufacturing

The pillars were made of EPDM rubber (KEP960N, Kumho Polychem, Seoul, Republic of Korea), which is a hyperelastic material used for sealing windows, cars, and washing machine doors. Figure 2a shows the shape of the pillar. Two types of cylindrical pillars with outer diameters of 3 mm (∅3 pillar) and 5 mm (∅5 pillar) and a height of 10 mm were manufactured. The ∅3 pillar was labeled as type A, and the ∅5 pillar was labeled as type B. There was a hole with a diameter of 1 mm at the center of the cylindrical pillar, through which the optical signal emitted by the phosphor attached to the end of the pillar spread to the camera.

Figure 2b shows the shape of the phosphor along with the excitation and emission wavelengths. A glass bead (HCMS-P-SLGS-FMR 2 mm, Cospheric, Goleta, CA, USA) made of soda–lime glass and coated with red fluorescence on both hemispheres was used as the phosphor. The excitation wavelength of the phosphor was 575 nm, and the emission wavelength was 607 nm.

Figure 2c shows the phosphor attached to the pillar. The phosphor and pillar were attached with an instant adhesive (Loctite460, Henkel Loctite Co., Ltd., Shandong, China). When used in the experiment, the pillar was mounted on a slide glass (76 mm × 26 mm × 1 mm; HSU-1000412, Marienfeld, Lauda-Königshofen, Germany). The same adhesive was used to fix the pillar to the slide glass, and the adhesive did not generate efflorescence that interfered with the optical signal because of the fast reaction time.

### 2.3. Experimental Setup

Figure 3 shows the optical measurement system used to measure the change in the optical signal when an external force is applied and the fluorescence image obtained from the optical measurement system. Figure 3a shows a photograph of the optical measurement system, which is located inside a tabletop darkroom to minimize interference from the external environment and includes equipment for applying the external force. A push–pull force gauge sensor (ISF-DF5A, Insize Co., Ltd., Suzhou New District, China) that can measure a maximum force of 5 N and has a resolution of 0.0005 N is employed to control the external force. This sensor is installed on an XYZ stage (#8095 Multi-Axis Piezo Device Alignment Stages, New Focus, Liwan District, China), which has an actuator that can move along three axes. The actuator can travel up to 3 mm, and its minimum incremental motion is lower than 30 nm. The inset (orange square) shows an enlarged view, which indicates that it is possible to find the pillar with the phosphor attached by using the push–pull force gauge sensor and applying an external force to the pillar.

Figure 3b shows the entire experimental setup. A USB camera (acA2440-75um, Basler, Ahrensburg, Germany), which is a monochrome type with a Sony IMX250 image sensor, is used to collect the fluorescence signal. The pixel size is 3.45 μm × 3.45 μm, and the sensing area is 8.45 mm × 7.07 mm. The camera has a telecentric lens (CompactTL™ Telecentric Lens, Edmund Optics, Barrington, NJ, USA) and optical filter (Bandpass filter, Edmund Optics, Barrington, NJ, USA). The telecentric lens minimizes focus dispersion caused by depth variations in the phosphor, and the optical filter blocks light from the light-emitting diode (LED) illuminant to ensure that only the wavelength of 607 ± 36 nm emitted from the phosphor is delivered to the camera. In addition, a ring-shaped LED (SpecBrightTM, Prophotonix, Boston, MA, USA) is employed to provide uniform light to the phosphor while minimizing the system volume. This illuminant emits light with a wavelength of 470 ± 10 nm. A slide glass is placed underneath the optical measurement system, and a pillar with a phosphor attached to the bottom of the slide is fixed. An LED excites the phosphor at the end of the pillar, and the phosphor emits light with a wavelength of 607 ± 36 nm. The emitted light passes through an optical filter to a USB camera, which captures the fluorescent image to analyze the magnitude and direction of the force applied to the pillar. The fluorescence signal is acquired as a circular image, because the camera is focused on the mating surface of the slide glass and pillar.

Figure 3c shows the fluorescent image that is altered under an applied external force. The brightness at 0.0 N decreases with a stronger external force. We use the average brightness value for image analysis. The average brightness value across 20,163 pixels within the yellow rectangular area is used, obtaining average values of 108 at an external force of 0.0 N, 80 at 0.9 N, and 56 at 1.3 N. Hence, it is possible to quantitatively convert the intensity of the fluorescence signal. The images are analyzed using Image J (Version 1.54 k), an open-source image processing software developed by the National Institutes of Health (NIH), USA. However, the system needs to be configured such that the brightest value (maximum) of any pixel does not exceed 255.

### 2.4. Simulation Conditions

The finite element method (FEM) based on fluid–structure interaction (FSI) was used to analyze the hyperelastic properties of the EPDM rubber in coupled physics, where fluidics and solid mechanics are combined. The FSI-FEM numerical analysis environment was built in COMSOL Multiphysics. Setting the properties of the materials used for numerical analysis is necessary, because the proposed sensor has a pillar structure with a phosphor attached. The glass bead used as a phosphor was assumed to be Pyrex 7740 glass. This glass has a density of 2230 kg/m^3^, Young’s modulus of 64 GPa, and Poisson’s ratio of 0.2.

The hyperelastic material properties of EPDM rubber have been analyzed in [36,37,38]; however, the property values vary depending on the composition and manufacturing conditions. Thus, we conducted a tensile test to obtain accurate values. The tensile test was performed by the Korea Testing & Research Institute using the KS M 6518 standard test method. As shown in Figure 4a, a dumbbell type three specimen was manufactured, and its hardness, strength, and elongation were 74, 15.2 MPa, and 270%, respectively, under tensioning at 500 mm/min. Figure 4b indicates the stress intensity–strain ratio curve of the EPDM rubber, and the coefficients of the Mooney–Rivlin nine-parameter model shown in Equation (1) were calculated as shown in Figure 4c. The results of fitting Figure 4b,c are presented in Appendix A. This model is appropriate for replicating the behavior of hyperelastic materials such as rubber and expressing the mechanical strain energy as a continuous sum of invariants [39].

The Mooney–Rivlin nine-parameter model can be expressed as follows:(1)$$W=C_{10}\left(I_{1}-3\right)+C_{01}\left(I_{2}-3\right)+C_{20}{\left(I_{1}-3\right)}^{2}+C_{11}\left(I_{1}-3\right)\left(I_{2}-3\right)+C_{02}{\left(I_{2}-3\right)}^{2}+C_{30}{\left(I_{1}-3\right)}^{3}+C_{21}{\left(I_{1}-3\right)}^{2}\left(I_{2}-3\right)+C_{12}\left(I_{1}-3\right){\left(I_{2}-3\right)}^{2}+C_{02}{\left(I_{2}-3\right)}^{3}$$

The flow field consisted of water with a density of 999.6 kg/m^3^ and dynamic viscosity of 0.001 Pa·s. These are the standard physical values (material library) provided by COMSOL. The flow fluid was set up assuming experimental environments; however, it could be different from the environment in which the pillar operates underwater. Figure 4d shows the designed chamber structure for the experiment. The interior of the chamber was 80 mm × 80 mm × 40 mm, and the distance from the hole where the fluid was injected was 6.2 mm. The nozzle connecting the tube to the chamber was designed to be tapered (Appendix A) to prevent backflow. The nozzle was simplified to facilitate the simulation with inner diameters of 7.86 and 6 mm. Further, it was modeled as a half-symmetric structure considering the symmetry of the target to be analyzed. The grating structure for the FEM analysis was created with tetrahedron elements, and the flow field consisted of 588,622 elements. In addition, we set a turbulent flow.

## 3. Results and Discussion

Figure 5 shows the simulation results from the FSI-FEM numerical analysis. Underwater flow is expressed in velocity units; however, most pumps are expressed in flow rate units. Therefore, the flow rate was converted into flow velocity through a simulation to develop an underwater sensor and analyze the displacement of the pillar in line with the change in flow velocity.

The flow field was analyzed to understand the behavior of the pillar with the attached phosphor. Figure 5a presents a vertical cross-sectional view of the flow field within the chamber. The flow of fluids occurring at the boundary layer was turbulent, and therefore, turbulent flow was set up by applying the Low Re k−ε model. The details of the boundary layer are shown in Appendix A. The simulation results indicated that the interior of the chamber did not show a uniform flow velocity distribution when a large flow rate was applied, and the fluid had a strong tendency to go straight to the opposite outlet. Therefore, the flow rate decreased as it moved away from the inlet. In Figure 5a, the 50 mm position is the inlet, and the −50 mm position is the outlet.

Figure 5b shows the change when flow rate is applied to the located pillar. The pillar within the chamber corresponds to the position of the red line in Figure 4d, indicating that the flow rate exerted a direct impact on the pillar. Figure 5c shows the relationship between the flow rate and velocity derived from the flow field experiments. Flow rates of 12, 60, 120, 240, 360, 480, and 600 mL/min were applied, and the flow velocities at the pillar locations were calculated through the simulation; the calculated flow velocities were 0.70, 5.33, 10.20, 19.18, 27.90, 36.53, and 45.08 cm/s, respectively. Based on these results, the first-order linear equation $y=7.462\times 10^{-2}x+0.7465$ was derived. This equation converted the flow rate into flow velocity. This equation can be applied to develop sensors made of EPDM rubber in an underwater environment.

Figure 5d shows that the simulated mechanical displacement of the pillar occurred by the flow field functioning on the pillar. The pillars were set to type A (∅3) and type B (∅5), and the converted flow velocity in Figure 5c was employed. The type A pillar showed deformations of 0.00, 0.01, 0.05, 0.21, 0.47, 0.84, and 1.32 µm, whereas the type B pillar showed deformations of 0.00, 0.00, 0.01, 0.05, 0.10, 0.19, and 0.29 µm. The equations used to represent the displacement of each pillar were $y=7.230\times 10^{-4}x^{2}-3.567\times 10^{-3}x+0.008583$ and $y=1.576\times 10^{-4}x^{2}-7.247\times 10^{-4}x+0.001638$ for types A and B, respectively. Based on these results, a flow rate of ~2650 mL/min was required to experiment with the displacement of the pillars at a flow velocity of ~2 m/s. The displacement of the type A pillar was 27.79 μm, and that of the type B pillar was 6.07 μm.

Figure 6 shows the results of analyzing the change in fluorescence signals when the force applied to the pillar changes. We performed the experiment by utilizing the optical measurement system in Figure 3, gradually increasing the force exerted on the pillar through the push–pull force gauge sensor and decreasing it again when it reached 0.18 N for the type A pillar and 1.8 N for the type B pillar. The fluorescence signal was measured when the increasing and decreasing external forces were of the same intensity, and this signal was calculated as the intensity by applying the analysis method in Figure 3c.

Figure 6a shows the results of the three repeated experiments with the type A pillar. A force up to 0.18 N was applied to the pillar, and the fluorescence signal was measured every 0.01 N. The fluorescence signal value at the initial state was 76.398 ± 0.281, and the intensity decreased with increasing force. When a force of 0.18 N was applied, the signal value was 12.518 ± 0.707, with slight differences of 13.135, 12.672, and 11.746 for each round. However, considering the initial signal values, no distinct tendency was observed. The signal of the pillar showed a hysteresis shape because of the nature of the pillar when the force decreased, and finally, the signal was 67.344 ± 0.106 when no force was applied, showing an average difference of 9.054 from the initial value. Figure 6b indicates the results of three repeated experiments with a type B pillar. A force was applied to the pillar up to 1.8 N, and the fluorescence signal was measured every 0.1 N. The signal value at the initial state was 76.325 ± 0.284, and the signal value was 18.230 ± 0.814 when a force of 1.8 N was applied, which was 5.712 higher than that of the type A pillar. The signal change appeared smooth up to 0.7 N, and it subsequently showed a radical change. Then, it became smooth again above 1.5 N. Further, it showed a hysteresis pattern when the force was reduced, although it was different from that of the type A pillar. Finally, the signal value was 70.400 ± 0.986 when no force was applied, showing an average difference of 5.925 from the initial value.

Although both pillar types exhibited similar initial signal values, a notable difference of 3.129 was observed in their final signal values after force application. This indicates that pillar B exhibited superior elastic recovery owing to its thicker, more rigid structure, which enhances its resistance to deformation and facilitates a more consistent return to its original shape. This enhanced resilience is particularly beneficial for real-world underwater applications, where maintaining measurement accuracy in dynamic flow conditions is essential. The elastic recovery could be further optimized by modifying the material composition or structural dimensions, potentially leading to the development of even more durable and resilient pillar designs. Regarding the measurement range, pillar A appears best suited for low-force detection, with an effective limit of approximately 0.1 N, whereas pillar B can accurately measure forces up to 1.8 N. However, to address intermediate force ranges, future sensor designs could focus on optimizing performance across a wider spectrum of force.

The fluorescence signal emitted from the pillar showed a difference from the signal in the initial state after applying a certain amount of force to the pillar and removing the force. This indicated the association with the elastic recovery/resilience feature of the pillar; therefore, it was necessary to confirm the recovery degree in line with the amount of force applied to the pillar. The occurrence of the elastic recovery indicated that strain energy remained inside the object. Figure 7 shows the results of measuring the elastic recovery degree of the pillar after a constant force was applied and removed. Three pillars were manufactured, and each pillar was subjected to three repeated experiments. The recovery level to the original state was analyzed by the difference in size between the initial signal value and the signal value after the experiment. The black square symbols represent the initial fluorescence signal, and the other symbols represent the fluorescence intensity measured after the experiment. The red circle, blue triangle, and green inverted triangle symbols represent the experimental results for the first, second, and third pillars, respectively, and the standard deviation was calculated based on the results of three repeated experiments.

Figure 7a shows the experimental result for the type A pillar. A force was applied in increments of 0.02 N from 0.02 to 0.18 N with the push–pull force gauge sensor. When a force of 0.02 N was applied, the fluorescence intensity decreased from 76.747 to 75.863 with an average difference of 0.884; when a force of 0.08 N was applied, it decreased from 76.663 to 74.143 with an average difference of 3.521; when a force of 0.18 N was applied, an average difference of 11.975 was found. The standard deviation was 0.172 up to 0.08 N because of experimenting with three pillars under the same conditions, which was similar to the standard deviation of the initial signal (0.186). However, the standard deviation of 0.276 was found from 0.1 to 0.14 N, and the standard deviations of 0.964 and 0.695 were observed at 0.16 and 0.18 N, respectively. Considering that the standard deviation increased with increasing force, the difference was not significant, and the reproducibility of the manufacturing method of the pillar seemed secured. Therefore, the elastic recovery feature was a material-related variable. Analyzing the results of three repeated experiments of each pillar indicated that the error between repeated experiments increased with increasing force. Further, the standard deviation increased rapidly from the point where the force of 0.1 N was applied. The average standard deviation at 0.18 N was 2.526, showing a very large error, which was attributed to the application of a large force that impeded the type A pillar to elastically recover. Thus, when using the type A pillar as a sensor, applying a force of 0.1 N or less was required for minimizing the error.

Figure 7b shows the experimental result of the type B pillar. A force was applied from 0.2 to 1.8 N at 0.2 N increments. When the force of 0.2 N was applied, the fluorescence intensity decreased from 76.227 to 75.788 with an average difference of 0.439; when the force of 0.8 N was applied, it decreased from 76.285 to 73.590 with an average difference of 2.696; and when the force of 1.8 N was applied, the average difference was 4.809. The result of experimenting with three pillars under the same conditions revealed that the measured values were similar, and the average standard deviation of the type B pillar was 0.193, which was half as low as that of the type A pillar. In addition, the standard deviation of the type B pillar did not increase even though the force increased, and the error between experiments averaged 0.240, which was comparatively lower than that of the type A pillar (0.761). Thus, the type B pillar showed better elastic recovery than the type A pillar. In conclusion, the displacement of the pillar and elastic recovery properties are inversely related, and the elastic recovery characteristics may depend not only on the material but also on the shape of the structure.

Although the degree of elastic recovery of the pillar varies depending on the size of the force, the experimental results in Figure 7 show that both type A and B pillars showed a linear change with increasing force. By analyzing this linear change, we suggest a formula to correct the elastic recovery features in Figure 6.

Figure 8a,b show the difference between the initial fluorescence intensity obtained when force was applied to the pillars in Figure 7a,b, respectively, and the fluorescence intensity after elastic recovery. Figure 8a shows the type A pillar, and the results follow the trend line of the first-order linear equation of y=67.486×x−1.539. Figure 8b shows the type B pillar, and the results follow the trend line of y=2.613×x+0.382. The slope of type A, which had a larger displacement in line with the applied force, changed more than that of type B. The Pearson’s *r* values for each trend line are 0.977 and 0.969, respectively.

Based on this linear equation, the equation for calibrating the elastic recovery characteristics is expressed as follows:(2)$$y=M+\left\{\left(a\times F\times \alpha \right)+b\right\},$$
where *M* represents the measured value, *a* and *b* represent the slope and intercept of each pillar, respectively, *F* represents the magnitude of the force previously applied to the pillar, and α represents a fitting constant determined by the material and shape. It is important to note that the sensor exhibits hysteresis, as evidenced by the response differences during force application and release. This hysteresis could potentially introduce inconsistencies in measurements, leading to drift or delays, particularly under dynamic flow conditions. Therefore, to ensure accurate flow measurements, it is crucial to calibrate the sensor and develop a hysteresis compensation mechanism. This will allow for correcting the sensor readings based on the specific loading history, thereby improving its reliability and accuracy in real-world applications.

Figure 8c shows the calibration results of Figure 6a obtained by applying the calibration equation to the type A pillar. In the three repeated experiments, the initial fluorescence signal values were 76.537, 76.074, and 76.582; after applying the force of 0.18 N, the final fluorescence signal values were 67.433, 67.373, and 67.227. The final values were substituted into *M*; the slope and intercept were substituted with 67.486 and −1.539, respectively; *F* was set to 0.17 N; and α was set to 0.92 to correct the force of 0.18 N. The final calibrated fluorescence signal values were 76.449, 76.389, and 76.243, respectively, which were similar to the initial values. The improvement rate was calculated as 99.585%.

The improvement rate was calculated using the following:(3)$$Improvement\;Rate=\left(\frac{Initial\;Error\;Mean-\left|Final\;Error\;Mean\right|}{Initial\;Error\;Mean}\right)\times 100.$$

Since this graph was corrected for all measured values at each force, the calibrated result showed a reduced area of the graph compared to that of Figure 6a. However, the shape of the graph remained the same.

Figure 8d shows the corrected result of Figure 6b with the calibration equation for the type B pillar. In the three repeated experiments, the initial fluorescence signal values were 76.435, 76.537, and 76.002, and after applying the force of 1.8 N, the final fluorescence signal values were 69.611, 71.506, and 70.083. The final values were substituted into *M*; the slope and intercept were substituted with 2.613 and 0.382, respectively; *F* was set to 1.7 N; and α was set to 1.25 to correct the force of 1.8 N. The final corrected fluorescence signal values were 75.546, 77.441, and 76.018, respectively, which was a substantial improvement of −0.010 compared with the existing error mean of 5.925. The improvement rate of the fluorescence signal was calculated to be 99.825%.

Figure 8d shows that the measured values at each force were all calibrated. Compared with Figure 6b, the area of the graph was reduced; however, the shape of the graph remained the same. Overall, the proposed correction method can improve the difference between the initial fluorescence intensity and fluorescence intensity after elastic recovery.

## 4. Conclusions

This study presented the development and initial validation of a novel biomimetic pillar-based flow sensor for detecting underwater flow by measuring elastic response and fluorescence signal changes. Two pillar types (A and B) were fabricated and characterized. The results indicated that pillar B, with its thicker and more rigid design, exhibited superior elastic recovery and resilience to deformation, consistently returning to its original shape after force application. This suggests that pillar B is a promising candidate for flow sensing in dynamic underwater environments, where maintaining measurement accuracy is critical.

Additionally, a method for quantifying fluorescence signal changes in response to the applied force was established, providing a foundation for future adaptation of the sensor for flow velocity measurements. The tensile properties of the EPDM rubber used to fabricate the pillars were analyzed, confirming its suitability for aquatic applications owing to its excellent durability and flexibility. Furthermore, a theoretical relationship between flow rate and velocity was derived, offering a potential pathway for future sensor calibration in real-world flow conditions.

However, this study primarily focused on establishing and validating the working principles of the proposed pillar-based sensor and characterizing its response to applied forces, including its elastic recovery characteristics. Therefore, a comprehensive comparison with existing flow or force sensors based on metrics such as measurement range, flow sensitivity, and maintenance requirements were not conducted, as this was early-stage research. In a future work, we will prioritize calibrating the sensor for specific flow measurement applications and benchmarking its performance against conventional flow and force sensors to assess its practical suitability. Additionally, its long-term reliability, including light source stability and expected lifespan of the phosphorescent material in underwater environments, will be investigated.

## Figures and Tables

**Figure 1 biomimetics-09-00721-f001:**
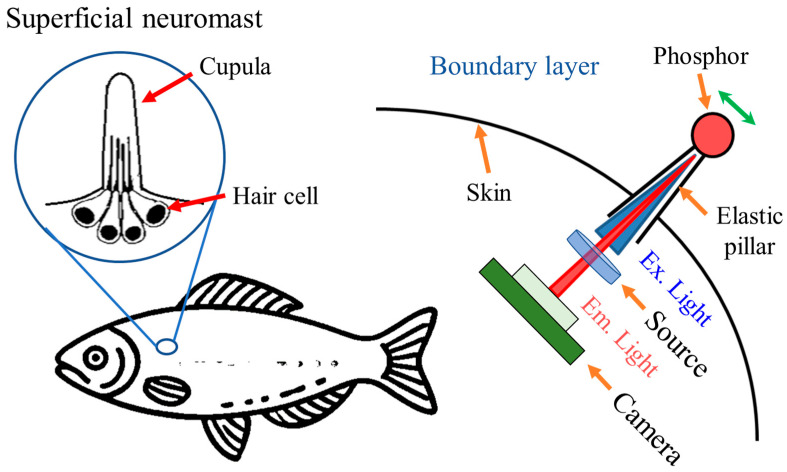
Fluid flow measurement sensor based on optical sensing and mimicking the superficial neuromast in fish. A pillar-shaped structure is used to induce mechanical deformation in response to the fluid flow. Unlike how fish generate electrical signals through mechanical deformation of the neuromast, this sensor measures fluid flow by analyzing the fluorescent signal of the phosphor generated by the mechanical deformation (indicated by the green arrow) of the pillar.

**Figure 2 biomimetics-09-00721-f002:**
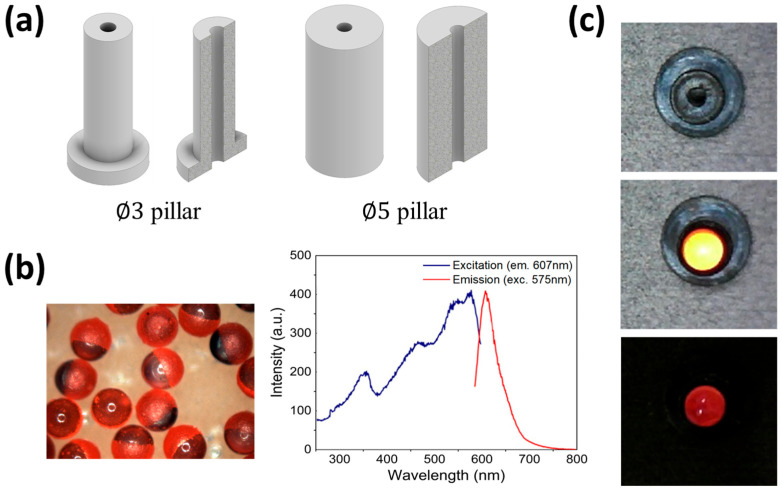
Designed form of pillar mechanically deformed by fluid flow in the boundary layer. Phosphor is used as a signal source. (**a**) The pillar is cylindrical with a hole of 1 mm in diameter. Two types of pillars, with outer diameters of 3 mm (type A) and 5 mm (type B), were manufactured and used in experiments. (**b**) Phosphor as a transparent bead made of glass and coated with red fluorescence on the surface of the hemisphere. The graph shows the excitation and emission wavelengths of phosphor. In this experiment, phosphor was excited with a wavelength of 470 ± 10 nm, and a bandpass filter was utilized with a wavelength of 607 ± 36 nm to selectively acquire the emitted fluorescence signal. (**c**) Attachment of phosphor to the pillar, showing the actual appearance of the type A pillar from the top, appearance after attaching the phosphor, and red fluorescence signal emitted from phosphor.

**Figure 3 biomimetics-09-00721-f003:**
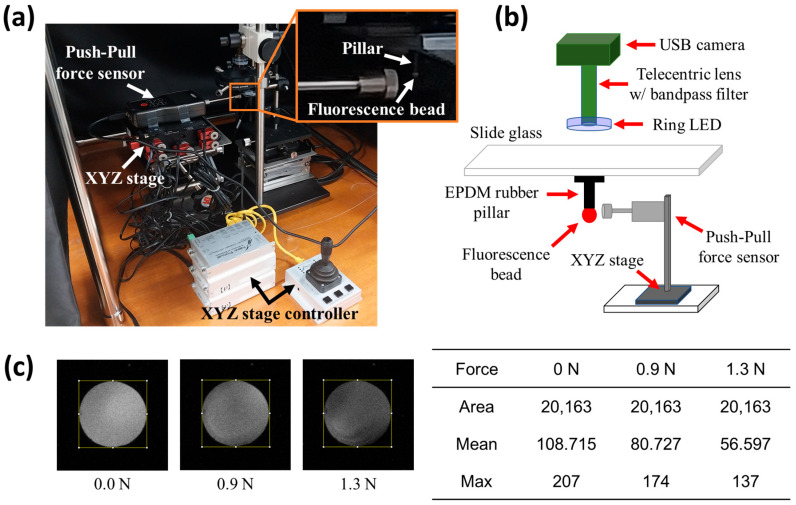
Optical measurement system for measuring the fluorescence signal of phosphors and analysis method of images obtained from the system. (**a**) Optical measurement system and external force application system for experiment. A push–pull force gauge sensor allows for quantifying the applied external force and is attached to the XYZ stage for precise movement. The enlarged area in the orange box shows the setup of the phosphor-attached pillar and push–pull force gauge sensor. (**b**) Diagram of experimental setup. The USB camera selectively obtains the fluorescence signal emitted by the phosphor and converts it into an image. (**c**) Fluorescence image and its analysis. The brightness of the fluorescence image decreases with increasing external force on the pillar. For the analysis, some areas (yellow square box) are selected from the fluorescence image, and the intensity of the pixels in the corresponding area is averaged.

**Figure 4 biomimetics-09-00721-f004:**
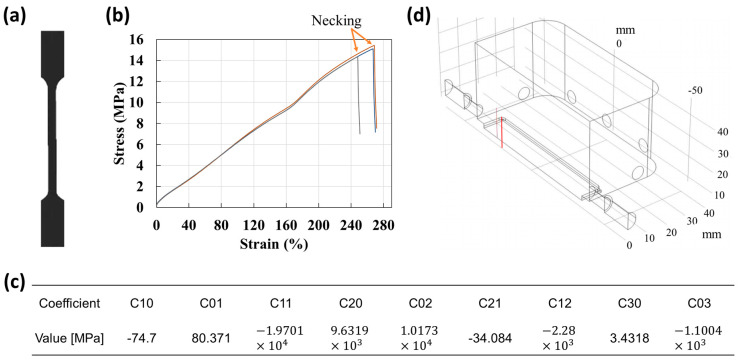
Property value derivation and simulation setup environment for simulating hyperelastic substances. (**a**) Specimen model for tensile test. The dumbbell type three specimen follows the guidance of the KS standard test method, KS M 6518. (**b**) Stress intensity–strain ratio curve. As a result of the test, the hardness, strength, and elongation were 74, 15.2 MPa, and 270%, respectively. (**c**) Parameters of the Mooney–Rivlin model derived from the stress intensity–strain ratio curve employing the ninth model. (**d**) Shape of chamber in which the flow field was analyzed. Considering the symmetry of the target to be analyzed, the pillar was modeled as a half-symmetric structure.

**Figure 5 biomimetics-09-00721-f005:**
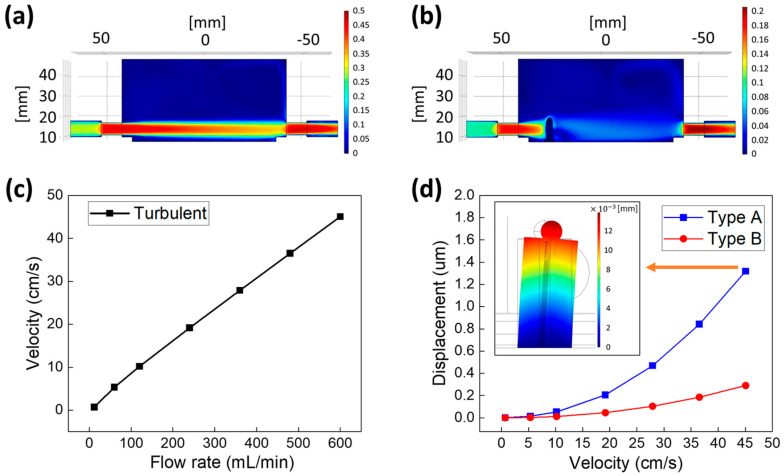
Flow field analysis of chamber using COMSOL Multiphysics and simulation of mechanical displacement of pillars in line with the applied flow velocity. (**a**) Simulation of flow field applied to chamber. (**b**) Simulation results of the flow field with pillars in the chamber. (**c**) Simulation results for converting applied flow rate into flow velocity, from which the linear function can be derived. It is possible to calculate the flow velocity on the pillar based on the flow rate setting in the chamber. (**d**) Simulation results of mechanical displacement of pillar in line with applied flow velocity. The pillar shape determines the pattern of the mechanical displacement. The image in the graph shows the mechanical deformation of the pillar when a flow velocity of 45 cm/s is applied to a type A pillar.

**Figure 6 biomimetics-09-00721-f006:**
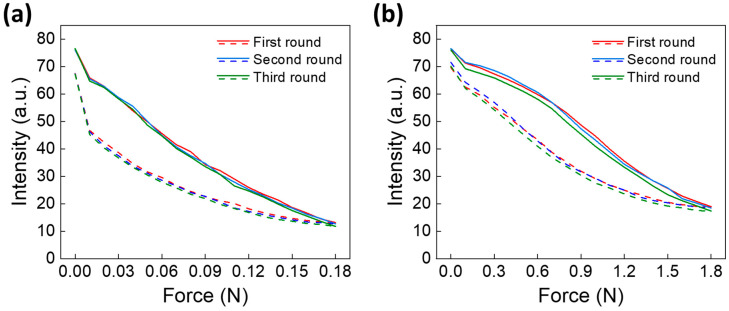
Fluorescence signal analysis results of pillars in line with applied force. (**a**) Experimental results for type A pillars. (**b**) Experimental results for type B pillars. Each experiment was repeatedly performed with the same pillar three times. The fluorescence signal was measured while increasing the force applied to the pillar and decreasing the force. The conditions of the optical measurement system were adjusted to match the initial intensity of both types. The solid and dashed lines represent the signal changes for increases and decreases in the force, respectively.

**Figure 7 biomimetics-09-00721-f007:**
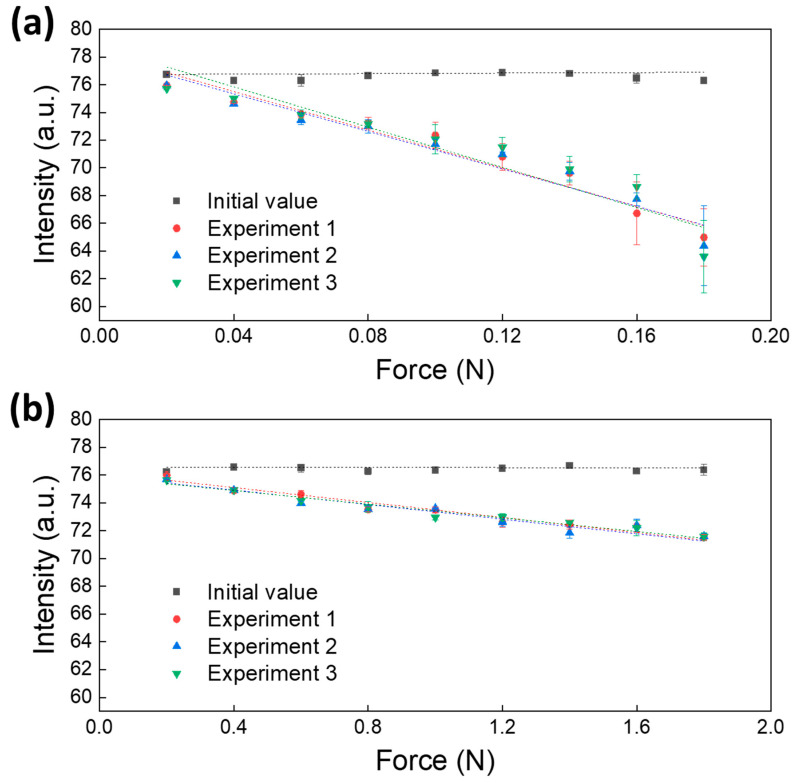
Quantitative analysis results of the applied force and resulting elastic recovery of pillars. (**a**) Experimental results for type A pillars. (**b**) Experimental results for type B pillars. Three pillars were manufactured, and each was utilized for three repeated experiments. The black square symbols represent the initial fluorescence intensity of the pillar, whereas the other symbols indicate the fluorescence intensity emitted after a constant force was applied to the pillar and then removed. The closer the measured fluorescence intensity is to the black square symbol, the greater is the elastic recovery property of the pillar.

**Figure 8 biomimetics-09-00721-f008:**
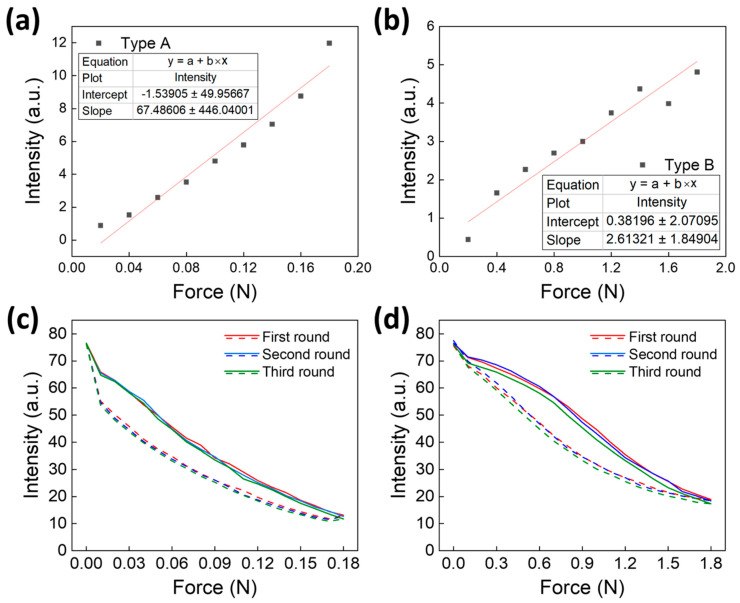
Method for correcting elastic recovery characteristics. (**a**) Intensities extracted from Figure 7a. (**b**) Intensities extracted from Figure 7b. Each intensity was extracted as the difference between the initial intensity and intensity after elastic recovery. (**c**) Corrected results for Figure 6a. (**d**) Corrected results for Figure 6b. After calibration, the overall area is reduced; however, the shape of the graph remains the same.

## Data Availability

Data underlying the results presented in this paper are not publicly available at this time but may be obtained from the authors upon reasonable request.

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
