# Peer review of "Optical Flow Sensor with Fluorescent-Conjugated Hyperelastic Pillar: A Biomimetic Approach"

_biomimetics, 2024, doi:10.3390/biomimetics9120721_

Round 1
Reviewer 1 Report
Comments and Suggestions for Authors
In this manuscript, the authors propose a sensor for water flow monitoring based on a fluorescent-conjugated hyperelastic pillar. The authors proposed two designs for two pillars capable of measuring different forces. Their results are solid, but there are some aspects fo the paper that need to be improved. Following, a series of comments aimed at enhancing the quality of the manuscript are included:
1. The authors indicate in the title that the sensor is able to measure water flow. Nevertheless, most of the results are given as the sensor's response to different forces instead of water flow. Thus, they have to change the title or change the way in which results are given. If the sensor is measuring water flow, it should be volume per unit of time.
2. In the second paragraph of the introduction, the authors should include a new approach for water flow monitoring based on the variation of their recently published optical properties: (2024). Proposal for Low-Cost Optical Sensor for Measuring Flow Velocities in Aquatic Environments. Sensors, 24(21), 6868.
3. More references need to be added, or detailed use of [12 to 16] must be made in the third paragraph of the introduction.
4. At the end of the introduction, the authors can include a short paragraph describing the structure of the rest of the paper.
5. In Figure 6 and similar cases, the authors must indicate which results correspond to increasing force and which to decreasing force. The authors must analyze how this hysteresis will affect the measuring conditions in real scenarios.
6. The authors have to explain why pillar B showed better elastic recovery and how this can be significant for the real deployment of the sensor. Is it possible to have a pillar with better elastic recovery? Will these sensors be able to measure the same forces? Where is the limit to using the A pillar and changing to the B pillar? Is it necessary to generate a new sensor to provide reliable data between the A and B pillars?
7. A comparison between the performance of the proposed sensors and existing flow sensors (or force sensors) must be provided at the end of the results. The authors have to highlight the differences in the performance of the sensors caused by different measuring ranges, flow velocities, and cost or maintenance needs.
8. The conclusions section must be summarized.
Author Response
We appreciate your invaluable feedback and have accordingly made the corresponding revisions in our manuscript. The changes are presented in red in the revised manuscript.
Please see the attached file for our point-by-point responses to your valuable comments, thank you very much.

Reviewer 2 Report
Comments and Suggestions for Authors
The article introduces a bioinspired optical flow velocity sensor designed to measure water flow accurately. The sensor utilizes phosphor attached to pillar structures, capturing fluorescence images with a USB camera to assess flow velocity through fluorescence intensity analysis. Two pillar types, A and B, were fabricated and tested for displacement and elastic recovery. A calibration method for adjusting elastic recovery across different pillar types was also proposed, establishing a foundation for bioinspired optical flow sensors.
The article is technically sound and well-structured. However, I have a few questions:
1. What is the expected lifespan of the phosphorescent material? Can it reliably support long-term measurements as described?
2. How is the quantity of phosphor controlled when it is attached to the transparent bead? Additionally, how does the material quantity influence fluorescence intensity?
3. How long was the testing period, and was the light source output stable over that duration? Could fluctuations in the light source significantly impact the measurement results?
Author Response

(The authors gave the same response as above.)

Round 2
Reviewer 1 Report
Comments and Suggestions for Authors
The authors have solved all the issues detected in the review, and I have no further comments.